# How Can a Polymeric Formula Induce Remission in Crohn’s Disease Patients?

**DOI:** 10.3390/ijms22084025

**Published:** 2021-04-14

**Authors:** Kawthar Boumessid, Frederick Barreau, Emmanuel Mas

**Affiliations:** 1INSERM, INRAE, ENVT, Université de Toulouse, UPS, F-31000 Toulouse, France; kawthar.boumessid@inserm.fr; 2Unité de Gastroentérologie, Hépatologie, Nutrition, Diabétologie et Maladies Héréditaires du Métabolisme, Hôpital des Enfants, CHU de Toulouse, F-31300 Toulouse, France

**Keywords:** Crohn’s disease, inflammatory bowel disease, exclusive enteral nutrition, mucosal healing

## Abstract

Crohn’s disease is an inflammatory bowel disease whose prevalence is increasing worldwide. Among medical strategies, dietary therapy with exclusive enteral nutrition is recommended as a first-line option, at least for children, because it induces clinical remission and mucosal healing. Modulen^®^, a polymeric TGF-β2 enriched formula, has good palatability and is widely used. For the first time in the literature, this review outlines and discusses the clinical outcomes obtained with this therapy, as well as the potential mechanisms of action of its compounds. It can be explained by its TGF-β2 content, but also by its protein and lipid composition. Further well-designed studies are required to improve our knowledge and to optimize therapeutic strategies.

## 1. Search Strategy

References for this review were identified on PubMed, Cochrane library, and other medical and dietary websites (Nestlé^®^, European food safety authority, dietary guidelines, clinical trials). The terms “Modulen”, “Polymeric”, “enteral nutrition”, “exclusive enteral nutrition”, “Crohn’s disease”, “Inflammatory bowel disease”, “mucosal healing”, “steroids” were used from 1994 until 2020. Articles indicating the use of the Modulen^®^ formula have been selected to discuss clinical remission and are summarized in Table 1. Concerning mechanisms of action, components have been discussed as well as other plausible compounds.

## 2. Introduction

### Crohn’s Disease

With 6.8 million cases in 2017 and an increasing worldwide prevalence of 85.1% from 1990 to 2017, inflammatory bowel disease (IBD) represents a conducive risk to issues in health, social, and economic systems [14]. Nowadays, the prevalence for Crohn’s disease (CD) in North America is 319 per 100,000 persons and 322 per 100,000 persons in Europe [15]. This spectrum combines CD and ulcerative colitis, both characterized by chronic intestinal inflammation. CD can affect the whole intestinal tract, from the mouth to the anus and the lesions are patchy and transmural. The complexity of the disease principally resides in its genetic and environmental causes. Among the 37 alleles specific to CD, the majority is related to immune reaction (nucleotide-binding oligomerization domain 2 (*NOD2*), *ATG16L1*…) or mucus layer (*MUC2*) [16]. The mutation of *NOD2* was one of the first characterized; this gene encodes for the pattern recognition receptor NOD2, described to regulate intestinal homeostasis. *MUC2* gene encodes mucin 2 secreted by Goblet cells to be part of the protective mucosal layer. Concerning environmental factors, cigarette smoking, antibiotic use, and a high saturated fat/low fiber diet are the main ones correlated to developing CD. In a general manner, the pathophysiology of CD is described as the outcome of an abnormal immune response, stimulated by intestinal microbiota dysbiosis. This last one consists of Gammaproteobacteria and Actinobacteria rise, as well as Bacteroides and Firmicutes decline. Even though causative explanations between microbiota and the immune system are not brought to light yet, the process is supported by intestinal permeability (IP) alterations. Indeed, increased IP along with tight junction protein modifications enhances luminal passage and thus immune stimulations. The resulting intestinal epithelium destruction displays, in turn, consequences on luminal content. A vicious circle is thus established, bringing difficulties to manage medical care. Hence, the totality of intestinal barrier compartments is disrupted: intestinal microbiota, mucosal layer, intestinal epithelium, and intestinal lymphoid tissue. Historically, the treatment of CD was based on anti-inflammatory drugs, i.e., corticosteroids to block the acute phase of inflammation, and on immunosuppressive drugs, like azathioprine and methotrexate, to prevent flare-up. However, although corticosteroids can reduce gastrointestinal symptoms, they have a low efficacy to achieve mucosal healing; 50% of CD patients failed to respond to corticosteroids [17]. Corticosteroids have also several side effects; they can, in particular, reduce growth velocity, which is often impaired by CD itself in children. Later on, biologics were developed to target specific inflammatory cytokines involved in IBD, like TNFα, p40 subunit of IL-12 and IL-23, and to block lymphocyte recruitment like α4β7 integrin antibody. However, up to 40% of IBD patients are non-responders to tumor necrosis factor (TNF) antagonists [18]. In order to have an appropriate drug approach, it is useful to keep in mind the mechanisms of drug resistance in IBD [19] and to optimize biologic treatments [20].

Besides the drug approach, nutritional therapy by exclusive enteral nutrition (EEN) or parenteral exclusive nutrition is applied in case of inadequate caloric intake, intestinal complications, and surgery. Nowadays, gastrointestinal pediatricians prefer to use EEN rather than corticosteroids in CD children because of its efficacy in mucosal healing and growth improvement. Unlike steroids, EEN improves children’s growth [21]. EEN has also the advantage to allow sufficient quantities of macronutrients and micronutrients, as well as qualitative consumption without harmful chemicals, improper cooking, and processing consequences [22]. However, the administration route and compliance are often difficult to manage for patients. Moreover, new insights suggest that some EEN ingredients might be harmful [23]. Thus far, anti-inflammatory drugs are still largely used. However, interest in EEN mushroomed these last years, but remains unclear. In this review, we focus on Modulen IBD^®^, because it is widely used in France and other countries for CD and because several clinical studies have investigated its impact on CD remission.

## 3. Modulen^®^ IBD

Facing CD, there are two overriding challenges for physicians, including undernutrition, which is a common flare-up consequence among patients, and corticosteroid side effects, which can alter the growth of CD children. Modulen^®^, allowed on the market in 2001, has been established to respond to these challenges. It is a polymeric formula for enteral or oral exclusive nutrition specifically dedicated to CD patients [24].

### 3.1. Composition

Modulen^®^ is a liquid food for special medical purpose indicated during flare-ups of CD patients [24]. Its exclusive use guarantees complete nutritional intake in terms of carbohydrates, lipids, and proteins with 44%, 42%, and 14% of total energy intake respectively (Table 2). These proportions are quite in line with the European and North American dietary intake recommendations, even though the lipid fraction exceeds the upper bound of the reference intake range by seven percent [25,26]. However, a lipidic reference intake higher than 35% is not unhealthy, since it can take account of dietary patterns and recommendations could vary between countries [27]. Among the lipids, saturated fatty acids (SFA) are the most represented (Table 2) and can be considered as a high proportion, since the lowest consumption is the best, but half of them will lead to medium-chain fatty acids (46% of SFA), which are crucial for Crohn’s disease diet [24]. Moreover, unsaturated fatty acids are the lowest lipidic proportion (16% of monounsaturated (MUFA) and 10% of polyunsaturated (PUFA)) [24]. Comparing to the labelling reference intake, quantities per day of n-6 PUFA are 2.4 g higher and n-3 PUFA is 1.2 g lower. Even if these quantities are included in the reference range, it leads to an unbalanced n-6/n-3 PUFA ratio [28]. A total of 13 vitamins and 15 minerals are provided in significant quantities, but it is not the case for sodium, potassium, and fluorides. Choline, an essential nutrient, is also not provided in an adequate amount [24,29]. Modulen^®^ is lactose, fiber, and gluten-free. Thus, carbohydrates mainly consist of glucose and sucrose. Concerning proteins, this liquid diet is 100% casein-based [24]. Last but not least, one of the characteristics of Modulen^®^ is its transforming growth factor β2 (TGF-β2) richness, an immunoregulatory cytokine also found in human milk and other EEN formulas (Santactiv Digest) [30,31]. This composition is adequate for the CD condition, in light of the robust clinical results obtained with Modulen^®^.

### 3.2. Modulen^®^ Induces Clinical Remission

In order to understand CD clinical management, it could be helpful to keep in mind some medical definitions. To evaluate the CD course, the main activity disease indexes used are CD-activity index (CDAI) or paediatric CD activity index (PCDAI), and Harvey Bradshaw index. The score lessening refers to the clinical response. Clinical remission, however, is defined as the normalization of the activity index. At the macroscopic scale, endoscopic remission refers to a normal mucosal appearance. More precisely, the absence of visible ulcerations during endoscopy is called mucosal healing and represents the best predictive criteria of sustained remission, and thus the main clinical objective [32]. At the microscopic level, histological remission is possible, indicating a complete normalization of impaired mucosa, e.g., a deep remission.

Modulen^®^ effectiveness to induce remission has been shown in many studies (Table 1). These studies have several limitations, while they are mainly retrospective or with no randomized control, and the endpoints were clinical remission rather than mucosal healing. Among patients treated exclusively with Modulen^®^, 65% of patients (19/27 children) have displayed a PCDAI ≤ 15 after 6–8 weeks [1] and 79% (23/29 children) have reached clinical remission with a PCDAI ≤ 10 after 8 weeks of CT32I Nestlé^®^ formula treatment [2]. A similar rate (80%) has been described by Buchanan et al. (105/114 children), who have combined both oral and nasogastric administration [3]. The mode of administration depends on patients’ clinical status [33], but does not affect the clinical remission rate. After eight weeks of exclusive Modulen^®^, oral and nasogastric administration induces 75% and 85% of clinical remission respectively (PCDAI < 10), without any statistical difference between treatments (Table 2) [4]. Considering the severe corticosteroid side effects, numerous studies have already illustrated that exclusive nutrition has an equal efficiency to corticosteroids in children contrary in adults [34]. To our knowledge, only two studies comparing corticosteroids to exclusive nutrition have been performed by adopting the current Modulen^®^ formula [5,6]. In the first one [5], an exclusive 10-week diet induced clinical remission and the PCDAI reduction was similar to corticosteroid treatment. However, endoscopic and histological healing was achieved at 73% in the polymeric diet group (14/19 children), significantly higher compared to 40% in the corticosteroid group (6/15). In the second one [6], PCDAI significantly decreased after eight weeks of polymeric diet compared to corticosteroids, from the second week until the third. In a long-term follow-up on mesalamine maintenance, the remission rate was longer when the induction treatment was performed by the polymeric diet, as more than 80% of individuals were in remission one year after. Another study has compared corticosteroids, cyclosporine A and enteral nutrition (EN) with an elemental diet (Flexical, Mead Johnson) or a polymeric diet (Nestlé) (Table 1) [35]. After 8 weeks of treatments, among the three patients treated with the polymeric diet, two of them had an improved histological inflammation. This outcome was similar to the elemental diet (5/6) and cyclosporine A (6/9), while it did not improve on prednisolone (1/10). However, the number of TNFα secreting cells only decreased on cyclosporine A.

Besides clinical response, endoscopic and histological responses were also investigated (Table 1) [2,4,5,6]. The EEN treatment was effective to induce ileal and colonic endoscopic improvement as well as histological healing [2], and even mucosal healing was achieved at week 8 [4]. Only few articles highlight the capacity of EEN to reach mucosal healing [36]. While numerous studies have been performed on medicines’ efficacy in mucosal healing [36], only one reported that Modulen^®^ is more efficient to achieve mucosal healing than corticosteroids (Table 1) [9]. Although Modulen^®^ exclusive nutrition is discerned as reliable medical management, its efficacy depending on ulceration localization, i.e., ileal/ileocolonic vs. colonic, is still a matter of controversy. On one side, the nutritional support response was not affected by the disease activity site [3,4], specifically by comparing small bowel disease and colonic disease (Table 1) [3]. On the other side, individuals with ileal and ileocolonic disease displayed a higher remission rate and improved endoscopic and histological scores [10].

The inflammatory status of patients was evaluated in parallel. The clinical studies demonstrate that symptoms decrease along with inflammatory serum markers such as C-reactive protein (CRP) and erythrocyte sedimentation rate (ESR) levels [1,2,4,7,9], platelets [1,4], fibrinogen [7], and TNFα levels [2] (Table 1). Among the studies already mentioned, three of them reported a significant increase in albumin levels. Furthermore, these serological results were accompanied by reduced inflammation at the mucosal level. Ileal biopsies from CD patients pre- and post-EEN revealed a decrease of IL-1β and IFN-γ mRNA, whereas colonic ones only presented IL-1β and CXCL-8 mRNA reduction [2,37]. These results attest to the anti-inflammatory effects of Modulen^®^.

As mentioned above, the improvement of nutritional status is a primary focus. This one is commonly assessed by anthropometric parameters. More precisely, a significant amelioration of weight z-score [1,3], body weight [2,4,7], body mass index [1,2,3], as well as skinfold thickness, and arm circumferences [7] were described. Concerning weight gain, it seems that it is better during oral nutrition [4] (Table 1).

Exclusive nutrition can be difficult to achieve due to its lack of palatability and its partial prescription could be a potential approach. Besides, in light of the insights into food promoting inflammation, controlling qualitatively the food intake seems to be an interesting concept. In an elegant study [8], the authors have established a special CD exclusion diet (CDED) based on some compulsory foods, some allowed foods across time, and the exclusion of foods inducing inflammation or discomfort (Table 2). This CDED was accompanied by 50% of daily energy requirements as Modulen^®^ during the first six weeks, then by 25% until week 12, which is also called partial enteral nutrition (PEN). In contrast, other children received EEN for six weeks followed by a gradually free diet with 25% of Modulen^®^. The results support that the exclusive diet with CDED/PEN allows significantly greater remission and tolerance than EEN. The corticosteroid-free remission rate (PCDAI < 10), respectively at week 6 and week 12, were 75% and 75.6% for CDED/PEN combination and 59% and 45.1% for EEN. However, it should be noted that the clinical remission rate of EEN is the lowest found in the literature so far.

### 3.3. Potential Mechanisms by Which Modulen^®^ Promotes Intestinal Renewal

There may be several ways that could explain how this formula operates (Figure 1, left panel). The liquid form is a valuable feature, since it reduces bowel movements and allows it to rest. This goes in hand with the exclusivity of the diet that can reduce dietary antigens intake and the consequent immunologic response, an intensive outcome in CD patients [38]. Concerning the composition, the lack of lactose, fibers, and gluten permits intolerant or hypersensitive patients to undergo therapy (Figure 1 left panel). Even if lactose intake has not been correlated to disease exacerbation, patients have experienced symptomatic relief after a low fermentable, oligo-, di-, mono-saccharides, and polyols (FODMAP) diet [39]. Symptoms reduction has also been self-reported by patients on a gluten-free diet, while no clinical trial has been performed yet [39]. Because fibers are incompletely digested carbohydrates, residues are decreased and thus subsequent stool frequency (Figure 1, left panel). As lactose, fiber exclusion alone has not been studied [40]. Even if numerous exclusion diets were investigated, further clinical trials examining lactose, fibers, and gluten in IBD are required.

Modulen^®^ is different from other EN formulas mainly by its TGF-β2 amount (Figure 1 right panel) [32]. This cytokine has intestinal benefits such as promoting IgA production, regulating tight junction proteins, and preventing Goblet cell depletion [41]. Furthermore, stimulating intestinal cells with TGF-β2 has down-regulated CXCL-8, IL-6, and TNFα (Figure 1, right panel) [42]. This lessening concerns both macrophage cytokines [42] and transcriptional level modifications [42,43]. Moreover, depleting TGF-β signalling emphasizes weight loss and inflammation in a mouse model of colitis [43]. Other studies have shown the ability of TGF-β2 to prevent necrotizing enterocolitis [44] and mucositis [45]. Knowing that TGF-β is also involved during restitution of mucosal healing [41], remission outcomes obtained with Modulen^®^ could be principally explained by this cytokine (Figure 1, right panel). However, other components may play a potential role and should not be excluded. This is the case of protein and fatty acid contents that deserve interest.

The Modulen^®^ formula is casein-based. This protein is significant, since it can protect TGF-β2 from duodenal enzymatic degradation [7]. Potential beneficial effects may be due to the whole protein or to its derived peptides (Figure 1, right panel). In an ileitis model, macroscopic and microscopic lesions, and Goblet cell depletion were protected by β-casofensin [46]. The amino acid profile of casein proteins is principally rich in two essential amino acids and one non-essential [47]. The first one is leucine (from 69 to 108 mg/g), which promotes cryptidin-1 production by Paneth cells via Slc7a8 transporter [48]. The second one is lysine (from 49 to 67 mg/g), which has anti-inflammatory properties as demonstrated by the reduction of weight loss, disease index, and inflammatory cytokines in dextran sulfate sodium (DSS) induced colitis [49]. Finally, glutamic acid presents the highest concentration (from 218 to 239 mg/g). This amino acid has been widely studied in the intestine and it is recognized as a principal actor in intestinal integrity (Figure 1, right panel). Not only glutamic acid can regulate proliferative, apoptotic, and inflammatory cellular pathways, but also tight junction proteins [50]. Glutamic acid can act directly on proteins such as ERK1/2, STAT, and HSF, and indirectly by enhancing growth factors’ effects like EGF and TGF-α (Figure 1, right panel).

Some studies in the scientific literature discuss fatty acids’ benefits to the intestinal mucosa. For instance, a palmitic acid-enriched diet has promoted B lymphocyte proliferation, IgA production, and cellular proliferation after a 75% bowel resection (Figure 1, right panel) [51]. Even if the whole fatty acids content of Modulen^®^ is not specified, some of them spotlighted may contribute to clinical remission (Figure 1, right panel) [32]. Among them, medium-chain triglycerides (MCT), which include caproic, caprylic, capric, and lauric acid esterified, are digested and absorbed easier than long-chain triglycerides. In comparison, MCT are shorter carbon chain, more hydrophilic, and then does not require bile acids or cholecystokinin. Their absorption is passive and permits to gain portal system without chylomicron formation (Figure 1, right panel). MCT have shown their capacity to enhance intestinal mass and cellular proliferation at the proximal level [52], as well as villi length, crypts depth, and IgA production [53]. Additionally, studies have demonstrated that MCT can attenuate *Clostridium difficile*-induced inflammation [54]. More specific outcomes have been presented in in vitro studies with IPEC-J2 cells, in which caprylic acid enhanced the β-defensin 1/2 secretion [55] and capric acid attenuates the oxidation, IP, and cyclophosphamide-induced inflammation (Figure 1, right panel) [56].

Modulen^®^ dietary therapy provides essential fatty acids (Figure 1, right panel). The admitted anti-inflammatory properties of α-linolenic acid are permitted by docosahexaenoic and eicosapentaenoic acids, along with their derived mediators (resolvins, docosatrienes, neuroprotectins) [57]. Even if the α-linolenic acid quantity in Modulen^®^ is low, its absorption is optimized by soya lecithin [58]. While linoleic acid is often associated with inflammation, prostaglandins E ensuing its metabolism have declined TNFα and IL-1β serum levels [57]. Other derivatives may be valuable such as 15-hydroxyeicosatetraenoic acid, whose production by intestinal glial cells is defective in CD patients [59]. The authors have shown its impact on IP regulation via zonula occludens 1 (ZO-1) expression. Another interesting fatty derivative is 10-hydroxy-cis-12-octadecenoic acid, produced by the *Lactobacillus plantarum* of intestinal microbiota (Figure 1 right panel) [60]. On one hand, this microbial-derived peptide improves the intestinal barrier by increasing Occludin expression; on the other hand, it alleviates *Helicobacter pylori* infection by inhibiting the futalosine pathway (Figure 1, right panel) [60]. Other microbial-derived peptides may be increased by Modulen^®^ therapy. However, short-chain fatty acid production may be unlikely since the formula is fiber-free. Among butyrate-producer germs, *Faecalibacterium* and *Anaerostipes* are diminished while *Ruminococcus torques* is enriched (Figure 1, right panel) [36]. Additionally, the lessening of *Anaerostipes* and *Faecalibaterium* may be explained by the lack of lactose, since they are lactate-utilizing bacteria. Results about *Roseburia* vary [8,36], probably due to fructose malabsorption that differs between individuals. The microbiota was also enriched in *Clostridium symbiosum*, *Clostridium ruminantium*, *Ruminococcus gnaves,* and *Clostridium hathewayi* [36], in contrast to *Haemophilus*, *Veillonella,* and *Prevotella* [8]. Besides that, the Shannon Index and OTUs number increase demonstrate the enhancement of microbial diversity after EN (Figure 1, right panel) [36]. The bacterial composition after Modulen^®^’s intake has returned to its pre-therapy stage. This phenomenon is associated with a regular diet upturn [8] and has been related to why EN therapy may not persist in the long term. Nonetheless, CDED has retained the consequent microbial composition, associated with a successful remission rate [8]. It is then highly plausible that the resulted bacterial composition is playing a crucial role in inducing remission, and maintaining it is a considerable approach.

## 4. Future Directions

To date, while the majority of medical treatments target the immune cell compartment of the intestinal mucosa to attenuate inflammation, Modulen^®^ EEN leads to significant clinical remission, but also to significant mucosal healing, the most significant remission parameter by far, targeting the intestinal barrier. The fact that this formula can be orally administrated due to its palatability confers a greater tolerance and compliance for CD patients. The more compliant the patient is, the greater the remission is [4]. Compliance can be affected by different factors, such as age, gender, and even beliefs [61]. To facilitate EN, allowing a regular diet is an alternative. Even if it seems to not revoke the benefits of Modulen^®^, additionally, regular diet can depend on personal education, beliefs, habits, and temptations, and then influence in somehow the outcome. Therefore, regulated partial nutrition could be a preferable approach to a free diet to control efficacy over time [8]. Few studies aimed to assess the effectiveness of Modulen^®^ in the maintenance phase of CD. Modulen^®^ represented around 40% of daily caloric intake [11,12,13]. Another approach could be to perform cycles of Modulen^®^, 2 weeks of EEN every 8 weeks, rather than daily use; the recruitment of this protocol was completed, but the data are not published yet (ClinicalTrials.gov, NCT02201693).

Numerous studies have brought to the fore nutritional, anti-inflammatory, and regenerative Modulen^®^ properties among children, but future studies should investigate the adult case. Interestingly, remission rates are superior in newly diagnosed patients [1]. This highlights the relevance of EN and clinicians should reconsider its medical first requirement in CD treatment strategies. Furthermore, the dietary therapy even impacted CD complications. In one study, enterocutaneous fistula was diminished in 4/8 patients and completely closed in one of them [7]. The available data are not sufficient to conclude about Modulen^®^’s impact on fistula and other complications, since these are generally exclusion criteria of clinical trials. Further well-designed studies are required to improve knowledge and to optimize therapeutic strategies.

Even though studies are contradicting each other about Modulen^®^ efficiency on disease location [3,4,10], numerous clinicians noticed that at least one ileal damage is requested. The pathophysiological differences may explain this outcome. For instance, *NOD2* mutations are associated with CD with at least one lesion located in the small intestine. In view of its numerous intestinal functions, this receptor can play a significant role in EN mechanisms after microbiota recognition. This suggests that the possible modification of the intestinal microbiota by Modulen^®^ could be primordial to induce remission. Additional studies on intestinal consequent microbiota are encouraged, both in terms of taxonomy and time. Albeit the outcome could suggest enhancing the dietary therapy with prebiotics or probiotics, one should note that the contrary could appear if the ensuing microbiota variations are highly specific.

The Modulen^®^ composition is adequate to complete dietary intake recommendations except for choline and potassium. The first one is an acetylcholine and betaine precursor, while the second one represents an abundant electrolyte involved in fluid balance and muscle contraction regulation. Choline and potassium deficiencies may occur at the end of the Modulen^®^ EN course like carotenoids [12]. Indeed, plasma levels of lutein, lycopene, and β-carotene were decreased, maybe causing defective defensive antioxidant mechanisms. The lack of data on nutritional deficits should be completed. These side effects could be added to minor known non-lasting side effects. However, as reported in the literature until now, the composition appears to be well balanced for CD remission and permits avoiding health status aggravation. Even if some of the ingredients may display deleterious effects, it is just a question of equilibrium with advantageous components. Aside from nutritional status, the formula compounds are almost certainly actors in Modulen^®^ clinical efficacy. Recently, Svolos et al. proved that an exclusion diet (CD-TREAT), mimicking Modulen^®^ composition with solid foods, leads to similar clinical, inflammatory, and intestinal microbial outcomes than EEN [62].

Other than intestinal symptoms, CD can lead to extra-intestinal ones such as bone, skin, ocular, and thromboembolic complications. These events were not investigated after EN and more specifically Modulen^®^ therapy. This goes along with other organ consequences. The most alarming repercussion is steatosis as non-alcoholic fatty liver disease is common in IBD. Considering TGF-β2 content, and the risk of hepatic fibrosis, hepatologists could avoid Modulen^®^ therapy for CD patients.

Finally, in addition to gut integrity, Modulen^®^ elements will likely benefit other organs and physiological processes allowing a well-being stage. Naturally, physical ameliorations go along with mental ones, thus achieving an effective quality of life.

In conclusion, the nutritional therapy with Modulen^®^ is successful to enable clinical remission and probably mucosal healing. Nevertheless, the conducted studies remain scarce, and randomized controlled clinical trials are not the majority, including only a small number of patients. The therapeutic impacts on microbiota, nutritional status, and extra-intestinal symptoms are still lacking. To elucidate Modulen^®^ lasting clinical efficiency, side effects, and mechanisms of action, further investigations are required.

## Figures and Tables

**Figure 1 ijms-22-04025-f001:**
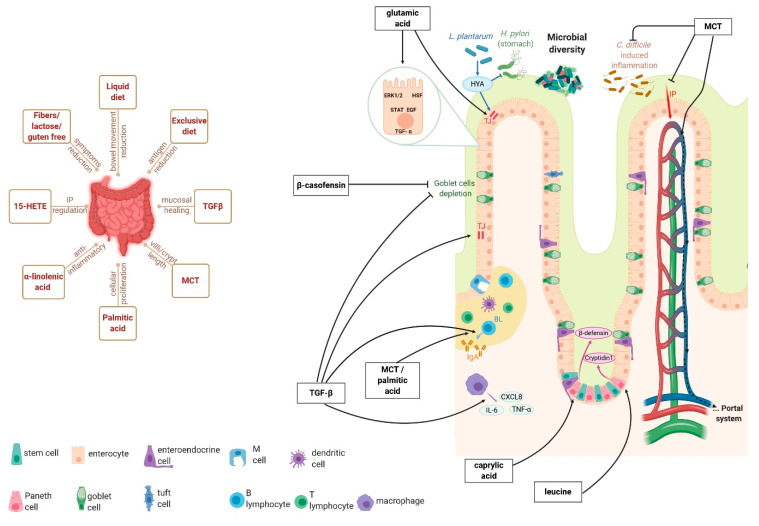
Mechanisms of action of Modulen^®^ on intestinal epithelium. The left panel represents the consequences of a liquid, exclusive, and fibers/lactose/gluten-free diet, as well as TGF(transforming growth factor-beta) MCT (medium-chain triglycerides), palmitic acid, α-linolenic acid, and 15-HETE (15-hydroxyeicosatetraenoic acid);. The right panel focuses on leucine, caprylic, glutamic and palmitic acid, MCT, TGF-β, and β-casofensin.

**Table 1 ijms-22-04025-t001:** Summary of the studies investigating Modulen^®^.

First Author	Study Type	Country	Administration (Route, Duration)	Number of Participants	Newly Diagnosed or Relapse	Evaluation Times	Endpoint
Day [1]	Retrospective	Australia	Oral ± NGT EEN/6–8 weeks	27 children	Both (15 newly diagnosed; 12 relapses)	8 weeks	Clinical remission: 80% (newly diagnosed) and 58% (long-standing)
Fell [2]	Pilot study	UK	oral (only one NGT) EEN/8 weeks	29 children	Both (17 newly diagnosed; 12 relapses)	8 weeks	Clinical remission: 79% Mucosal improvement: ileal (15/22) and colonic (13/26)
Buchanan [3]	Retrospective	UK	57 orally and 53 NGT EEN/8 weeks	110 children (105 with Modulen^®^)		8 weeks	Clinical remission: 80%
Rubio [4]	Retrospective	France	45 orally and 61 NGT EEN/8 weeks	106 children	Newly diagnosed or with a first relapse	8 weeks	Clinical remission: 75% (oral) and 85% (NGT)
Borrelli [5]	Open-label controlled trial	Italy	orally EEN/10 weeks	19 children	Newly diagnosed	10 weeks	Clinical remission: 79% Mucosal healing: 74%
Berni Canani [6]	Retrospective	Italy	Orally (12 Modulen) and NGT (13 semi-elemental, 12 elemental) EEN/8 weeks	37 children	Newly diagnosed	2/4/8 weeks	Clinical remission: 86.5% Mucosal healing: no difference between the 3 diets
Triantafillidis [7]	Pilot study	Greece	Orally 4 weeks EEN (medical treatment unchanged)	29 adults		4 weeks	Clinical remission: 38%; clinical improvement: 31%
Levine [8]	An open-label prospective randomized controlled trial	Canada and Israel	12 weeks EEN (34) or partial nutrition with CDED (40)	74 children		3/6/12 weeks	Clinical remission at W6: CDED = 75%, EEN = 58.8%
Pigneur [9]	Prospective randomized trial	France	8 weeks EEN	13 children	Newly diagnosed	8 weeks	Clinical remission: 100% Mucosal healing: 89%
Afzal [10]	Observational	UK	60 orally and 5 NGT 8 weeks EEN	65 children	Both (54 newly diagnosed; 11 relapse)	8 weeks	Clinical remission: 50% (colonic group), 82.1% (ileocolonic group), and 91.7% (ileal group)
Lionetti [11]	Pilot study	Italy	Orally 8 weeks EEN Maintenance EN	9 children	Newly diagnosed (7); relapse (2)	8 weeks	Clinical remission: 89%
Gerasimidis [12]	Pilot study	UK	Orally or NGT 6–8 weeks EEN Maintenance EN	17 children	Newly diagnosed or relapse	6–8 weeks	Clinical remission: 47%; Clinical response: 24%
Duncan [13]	Retrospective	UK	Orally (60%) and NGT (40%) 8 weeks EN Maintenance EN	59 children	Newly diagnosed	8 weeks	Clinical response/remission: 81%

**Table 2 ijms-22-04025-t002:** Composition of Modulen IBD^®^.

	100 g	Per 100 mL (1.0 Kcal/mL)		100 g	Per 100 mL (1.0 Kcal/mL)
**Energy (kcal)**	493	99	**Fats (g)**	23	4.6
**Carbohydrates (g)**	54	11	Saturated fatty acids (g)	13	2.6
**Proteins (g)**	17.5	3.5	Medium chain triglycerides (g)	6	1.2
**Minerals**			Monounsaturated fatty acids (g)	3.9	0.78
Sodium (mg)	170	34	Polyunsaturated fatty acids (g)	2.5	0.50
Potassium (mg)	600	120	-α linolenic acid (mg)	200	40
Chloride (mg)	365	73	-Linoleic acid (mg)	2100	420
Calcium (mg)	445	89	**Vitamins**		
Phosphorus (mg)	300	60	A (µg)	410	82
Magnesium (mg)	100	20	D (µg)	4.9	0.98
Iron (mg)	5.4	1.1	E (mg)	6.5	1.3
Zinc (mg)	4.7	0.94	K (µg)	27	5.4
Copper (mg)	0.49	0.098	C (mg)	47	9.4
Manganese (mg)	0.98	0.20	Thiamin (mg)	0.59	0.12
Fluoride (mg)	<0.10		Riboflavin (mg)	0.64	0.13
Selenium (µg)	17	3.4	Niacin (mg)	5.8	1.2
Chromium (µg)	25	5	B6 (mg)	0.83	0.17
Molybdenum (µg)	37	7.4	Folic acid (µg)	120	24
Iodine (µg)	49	9.8	B12 (µg)	1.6	0.32
**Other nutrients**			Biotin (µg)	16	3.2
Choline (mg)	35	7	Pantothenic acid (mg)	2.4	0.48
Osmolarity (mOsm/L)	290	290

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
