# Peer review of "How Can a Polymeric Formula Induce Remission in Crohn’s Disease Patients?"

_ijms, 2021, doi:10.3390/ijms22084025_

Round 1
Reviewer 1 Report
In my opinion there are also some other formulas in the market containing TGF-β (see Triantafillidis et al, Nutrients 2020 Apr 10;12(4):1048. doi: 10.3390/nu12041048.). Please mention these data
Please add a separate table with the ingredients of Modulen IBD
Author Response
In my opinion there are also some other formulas in the market containing TGF-β (see Triantafillidis et al, Nutrients 2020 Apr 10;12(4):1048. doi: 10.3390/nu12041048.). Please mention these data.
We thank reviewer 1 for his careful and positive examination of our manuscript.
In agreement with this concern, we have modified the introduction of the ModulenIBD paragraph to specify in the revised version of the manuscript that several formula containing TGF-b2 have been developed by the food industry. We have also add this data (Triantafillidis et al, Nutrients 2020 Apr 10;12(4):1048).
Please add a separate table with the ingredients of Modulen IBD.
In agreement a separate table including the main compounds of Modulen IBD has been performed and added in the revised version of the manuscript.
Reviewer 2 Report
The authors have provided a narrative review on the use of the use of polymeric exclusive enteral nutrition, specifically Modulen, for the treatment of Crohn’s disease. The article discusses evidence for efficacy of Modulen, it’s potential mechanism and future directions. There are a number of grammatical errors in the manuscript which do detract from the quality of the work. I have the following suggestions:
Do the authors believe that the specific type of polymeric EEN is important in the efficacy of therapy – that is, why was Modulen specifically chosen rather than polymeric EEN more broadly?
In section 3.2, it discusses the improved outcomes with endoscopic remission and mucosal healing but focuses on the use of clinical remission with Modulen. Consider doing separate sections on evidence for clinical remission and objective disease remission. The section on partial enteral nutrition should be discussed separately.
Is any data available on the dose and duration of therapy and whether this may have an impact on efficacy? If the focus is just on Modulen then further details should be provided on how clinicians can best use this therapy.
Minor points:
Introduction line 1 – an 85.7% increase in prevalence should be further clarified – what was the prevalence and what is it now?
Page 3 line 173 – the use of anecdotal evidence to support efficacy should be avoided, consider revising
Author Response
Rewiever 2
The authors have provided a narrative review on the use of the use of polymeric exclusive enteral nutrition, specifically Modulen, for the treatment of Crohn’s disease. The article discusses evidence for efficacy of Modulen, it’s potential mechanism and future directions. There are a number of grammatical errors in the manuscript which do detract from the quality of the work. I have the following suggestions:
We thank reviewer 2 for his careful review and remarks about our manuscript.
Do the authors believe that the specific type of polymeric EEN is important in the efficacy of therapy – that is, why was Modulen specifically chosen rather than polymeric EEN more broadly?
In general manner, we think that polymeric EEN is new and pertinent therapeutic strategy to treat CD. Among the numerous EEN formulas available, we have decided to focus on ModulenIBD because:
-this formula is used in numerous countries for CD treatment
-numerous clinical studies have investigated its impact on CD remission
-it is possible to use it orally
Thus, we think that focusing this review on ModulenIBD will bring a state of the art about the use of this formula as well as the role of its compounds that are potentially of interest to induce clinical remission.
All of these arguments have been added at the end of the introduction of the revised manuscript.
In section 3.2, it discusses the improved outcomes with endoscopic remission and mucosal healing but focuses on the use of clinical remission with Modulen. Consider doing separate sections on evidence for clinical remission and objective disease remission. The section on partial enteral nutrition should be discussed separately.
This is true because the studies performed on Modulen® investigated clinical remission rather than mucosal healing.
In order to be concise, we specified within the manuscript if it is clinical remission (rather than remission alone). However, data were already present on mucosal healing.
Finally, we moved data on partial enteral nutrition at the end of this section, because it is a quite new approach.
Is any data available on the dose and duration of therapy and whether this may have an impact on efficacy? If the focus is just on Modulen then further details should be provided on how clinicians can best use this therapy.
We thank the reviewer for this comment. The duration of EEN with Modulen® is 8 weeks in most studies reported in Table 1. The ECCO/ESPGHAN recommended a 6-8 weeks duration. The experts proposed to consider an alternative if EEN does not induce clinical response within 2 weeks. Finally, in the absence of data, they proposed at the end of 6-8 weeks EEN to reintroduce foods over 2-3 weeks.
Minor points:
Introduction line 1 – an 85.7% increase in prevalence should be further clarified – what was the prevalence and what is it now?
In agreement we have clarified this point. Nowadays the prevalence for Crohn disease in North America is 319 per 100000 persons and 322 per 100000 persons in Europe.
Page 3 line 173 – the use of anecdotal evidence to support efficacy should be avoided, consider revising
In agreement we have revised this sentence.
Reviewer 3 Report
Boumessid et al. is literature review outlining and discusses the clinical outcomes obtained with Modulen®, a polymeric TGF-β2 enriched formula therapy, as well as the potential mechanisms of action of its compounds and clinical remission are summarized. There are two overriding challenges for physicians include undernutrition, which is a common flare-up consequence among patients and corticosteroid side effects, which can alter growth of CD children.
The nutritional therapy with Modulen® is successful to enable clinical remission and even mucosal healing. Nevertheless, the conducted studies remain scarce and randomized controlled clinical trials are not the majority, including only small number of patients. The therapeutic impacts on microbiota, nutritional status, and extra-intestinal symptoms are still lacking. To elucidate Modulen® lasting clinical efficiency, side effects and mechanisms of action, further investigations are required.
To my view the available data are not sufficient to conclude about Modulen® impact on fistula and other CD complications since these are generally exclusion criteria of clinical trials. Further well-designed studies are required to improve knowledge and to optimize therapeutic strategies.
This is a very good and well summarized review. The Table 1 is informative. Citations are relevant.
Please check the language grammar such as on Line 95, there is a missing full stop after reference [10] and the like throughout the paper.
Author Response
Reviewer 3
Boumessid et al. is literature review outlining and discusses the clinical outcomes obtained with Modulen®, a polymeric TGF-β2 enriched formula therapy, as well as the potential mechanisms of action of its compounds and clinical remission are summarized. There are two overriding challenges for physicians include undernutrition, which is a common flare-up consequence among patients and corticosteroid side effects, which can alter growth of CD children.
The nutritional therapy with Modulen® is successful to enable clinical remission and even mucosal healing. Nevertheless, the conducted studies remain scarce and randomized controlled clinical trials are not the majority, including only small number of patients. The therapeutic impacts on microbiota, nutritional status, and extra-intestinal symptoms are still lacking. To elucidate Modulen® lasting clinical efficiency, side effects and mechanisms of action, further investigations are required.
To my view the available data are not sufficient to conclude about Modulen® impact on fistula and other CD complications since these are generally exclusion criteria of clinical trials. Further well-designed studies are required to improve knowledge and to optimize therapeutic strategies.
This is a very good and well summarized review. The Table 1 is informative. Citations are relevant.
Please check the language grammar such as on Line 95, there is a missing full stop after reference [10] and the like throughout the paper.
We thank reviewer 3 for his careful and positive examination of our manuscript. In agreement with his concerns, we have carefully checked the English language and grammar.
Round 2
Reviewer 2 Report
The authors have addressed the suggested comments adequately.
In the acronym for FODMAP, the F stands for "Fermentable" so please change this
Author Response
We thank reviewer 3 for his careful examination of our manuscript and in agreement with his concern, we have corrected the acronym for FODMAP.